# Eight Years of Real-Life Experience with Smoothened Inhibitors in a Swiss Tertiary Skin Referral Center

**DOI:** 10.3390/cancers14102496

**Published:** 2022-05-19

**Authors:** Lara E. Grossmann, Egle Ramelyte, Mirjam C. Nägeli, Reinhard Dummer

**Affiliations:** Department of Dermatology, University Hospital of Zurich, 8091 Zurich, Switzerland; egle.ramelyte@usz.ch (E.R.); mirjam.naegeli@usz.ch (M.C.N.); reinhard.dummer@usz.ch (R.D.)

**Keywords:** advanced basal cell carcinoma, hedgehog inhibitors, smoothened inhibitors, systemic treatment of basal cell carcinoma, real-world experience

## Abstract

**Simple Summary:**

Vismodegib and sonidegib are targeted therapies inhibiting the hedgehog pathway, a key driver in the pathogenesis of basal cell carcinoma (BCC). Hedgehog inhibitors (HhIs) are first-line therapy for locally advanced basal cell carcinoma (laBCC), metastatic basal cell carcinoma (mBCC) and multiple BCCs, when surgery and radiotherapy are no longer feasible. Safety and efficacy of the HhIs vismodegib and sonidegib have been shown in large prospective clinical trials. However, treatment of advanced basal cell carcinoma (aBCC) in daily practice includes patients who do not meet strict inclusion criteria and poses an additional challenge for treating physicians. This study aims to give an insight into a real-world experience in our tertiary skin referral center.

**Abstract:**

Background: The hedgehog inhibitors vismodegib and sonidegib are approved for the treatment of advanced basal cell carcinoma. This study reports the experiences with these therapies in a tertiary skin referral center in daily practice. Methods: A retrospective, observational, single-center study analyzing medical records of patients with aBCC treated with a smoothened (SMO) inhibitor outside a clinical trial for at least one month between 2013 and 2021. Results: In total, 33 patients were included: 21 (64%) patients were treated with vismodegib, 3 (9%) patients with sonidegib and 9 (27%) patients with both treatments subsequently. With vismodegib, the best overall response was complete response (CR) in 33% cases, and partial response (PR) in 33% cases. Under sonidegib, 42% patients achieved CR and 17% PR. Mean duration to next treatment was 33 and 14 months for vismodegib and sonidegib, respectively. Adverse events varied in frequency between continuous and intermittent dosing and they were the most common reason for therapy discontinuation. Conclusions: Our real-world data illustrate the pitfalls and benefits of HhIs as well as the impact of different dosing regimens on adverse events, patient adherence and response. Treatment duration remains limited by adverse events and resistance. Additional treatment options, including immunotherapy and drug combinations, are needed.

## 1. Introduction

Basal cell carcinoma (BCC) is the most common cancer among the white population and its incidence is rising worldwide [1,2]. It originates from long-term resident keratinocyte progenitor cells of the interfollicular epidermis and the upper infundibulum and is mainly caused by mutagenesis through chronic UV radiation [3,4]. Most BCCs grow slowly, rarely metastasize and can often be cured by surgery [5]. However, about 0.6% of patients suffer from advanced basal cell carcinoma characterized by extensive tissue invasion, distant metastasis or development of multiple recurrent BCCs within hereditary syndromes such as basal cell nevus syndrome (BCNS, also called Gorlin’s Syndrome) and Xeroderma pigmentosum. Advanced BCCs are no longer amenable to surgery or radiotherapy [6].

The HhIs or SMO inhibitors vismodegib and sonidegib have been approved for the treatment of advanced basal cell carcinoma in many countries [7,8,9,10]. Their mechanism of action involves the inhibition of the oncogenic protein smoothened, a key regulator of the hedgehog pathway. Mutations in this pathway leading to aberrant activation are found in most sporadic and BCNS-linked BCCs and play an important role in the pathogenesis and development of basal cell carcinoma [11].

Vismodegib was the first approved HhI for the treatment of advanced basal cell carcinoma based on the results of a multicenter international study (ERIVANCE), followed by sonidegib, which was approved based on safety and efficacy outcomes in the BOLT trial [12,13]. HhIs have since then become crucial in the management of these patients.

We initiated a retrospective analysis of aBCC patients treated with HhIs in order to investigate the benefits and limitations of HhIs, the tolerability in unselected patients and the management strategies to dampen adverse events. Some of our patients might qualify for the immunotherapy with cemiplimab outside of clinical trials.

## 2. Materials and Methods

In this retrospective single-center analysis, the Swiss hospital electronic database KISIM Version 5.3.1.5 was queried for adult patients who were treated with HhIs (vismodegib or sonidegib) for at least one month outside a clinical study. The time frame of the analysis was between drug approval of first HhI vismodegib in Switzerland on 1 October 2013 and 1 October 2021. Queries identified 36 patients, of which 33 were further investigated (Figure 1). Three patients were excluded due to missing informed consent.

Indication criteria for vismodegib and sonidegib included laBCC and mBCC ineligible for surgery and radiotherapy due to repeated recurrence after surgical procedures with curative intent or due to an expected considerable morbidity and deformity after surgery, or severe comorbidities. Patients with multiple BCCs (>5) including those with BCNS and Xeroderma pigmentosum were also eligible. All patients were discussed at an interdisciplinary tumor board prior to therapy start with a smoothened inhibitor.

We collected clinical data on age, gender, comorbidities, tumor location, previous therapies, indication for treatment with HhIs (laBCC, mBCC or multiple BCCs), type of HhI therapy, treatment dosage, dosing regimen (intermittent vs continuous), duration of intake, combination with other drugs, adverse events, management of adverse events, reason for therapy discontinuation and subsequent treatments.

We further obtained data on treatment response, which was clinically assessed by the investigator, including dermoscopy, according to the patient’s medical record in laBCC, and according to the RECIST criteria in patients with mBCC. For patients with multiple BCCs, response was also assessed clinically, including dermoscopy. CR was defined as disappearance of all lesions. PR was defined as clinical disappearance of some of the BCCs and surgical resection of residual BCCs that were clinically not responding according to the treating physician. Stable disease (SD) was defined as neither growing nor shrinking of all BCCs. Progressive disease (PD) for patients with multiple BCCs was defined as the formation of new BCCs under therapy (<3 months of treatment interruption) that were histologically confirmed. Patients who had an evaluable response were included in efficacy analysis (*n* = 25, 76%) and all patients were analyzed for safety.

## 3. Results

### 3.1. Study Population

A total of 36 patients had a treatment with a HhI, vismodegib 150 mg daily or sonidegib 200 mg daily, for at least one month outside a clinical study in our center between October 2013 and October 2021. Three patients had no informed consent for a retrospective data analysis and were excluded, leaving 33 patients from which 21 patients had monotherapy with vismodegib (64%), 3 patients with sonidegib (9%) and 9 patients (27%) had both drugs subsequently (Figure 1).

The mean age at treatment start of a HhI hedgehog inhibitor was 65.4 years (median 70 years, range 30–96 years) for all patients. Among the patients, 18 were men (55%) and 15 were women (45%). Most patients (*n* = 32, 97%) had at least one previous therapy, including surgery and radiotherapy as well as photodynamic therapy and systemic retinoids as chemoprophylaxis in nonmelanoma skin cancer. Six patients (18%) had prior therapy with a HhI within a clinical trial. The indication for vismodegib and sonidegib in this patient population was locally advanced BCC (*n* = 11, 33%), metastatic BCC (*n* = 5, 15%) and multiple BCCs (*n* = 17, 52%) including 8 patients with BCNS and 2 patients with Xeroderma pigmentosum. Most primary tumors of patients with laBCC and mBCC (*n* = 16) were located in the head region (*n* = 14, 88%), of which six were in the periorbital region, three on the scalp, two on the nose, two on the cheek and one in the maxillary sinus. Two patients (12%) had a primary tumor on the trunk. Our patient population represented a heterogenous group regarding comorbidities, such as most commonly cardiovascular disease (*n* = 14, 42%), osteoporosis (*n* = 3, 9%) and noncutaneous malignancies, including active hematologic malignancies and solid tumors (*n* = 3, 9%) as well as malignancies in their previous medical history (*n* = 2, 6%).

### 3.2. Dosing Regimen

#### 3.2.1. Vismodegib

Eleven patients (37%) had continuous therapy, eleven patients (37%) had intermittent dosing and eight patients (27%) had no specific dosing regimen with treatment cycles of varying duration alternating with treatment interruptions (Figure 1).

Four patients with continuous dosing had a therapy interruption lasting between 1–6 months, mostly due to adverse events, or in one case due to another acute malignancy. Mean duration of therapy was 17 months (range 1–62 months) without calculating treatment interruption.

From eleven patients with intermittent dosing, ten patients had a “2–3 months on/2–3 months off” schedule and one patient a “1 month on/one month off” schedule. Most of these patients (*n* = 7, 64%) had an induction period with continuous intake at the beginning of therapy ranging from 5–9 months. Some patients (*n* = 4, 36%) interrupted treatment for 1–6 months due to adverse events, temporary CR (with subsequent relapse) and nonadherence. Mean duration of treatment was 27 months (range 9–67 months), without calculating treatment interruption.

Eight patients, all with the indication of multiple BCCs, had no specific dosing regimen with treatment cycles of varying duration alternating with treatment interruptions up to 30 months. Mean treatment duration of this subgroup was 46 months with mean drug exposure of 4 months/year.

Looking at the subgroups, more than half of the patients with laBCC had continuous dosing (*n* = 5, 56%) and the others had intermittent dosing (*n* = 4, 44%). The majority of patients with mBCC had continuous therapy (*n* = 4, 80%) and one patient (*n* = 1, 20%) intermittent therapy. Most patients with multiple BCCs had no specific dosing schedule (*n* = 8, 50%), six patients (*n* = 6, 37.5%) had intermittent and two patients (*n* = 2, 12.5%) continuous treatment.

#### 3.2.2. Sonidegib

Half of patients (*n* = 6, 50%) had continuous therapy with two patients undergoing therapy interruption for 1 and 4 months, respectively, due to adverse events. Mean duration of intake was 10 months without calculating therapy interruption. The other half of patients (*n* = 6, 50%) had an intermittent therapy with an on/off interval of 2–3 months except for one patient with drug intake every other day. One patient had a previous induction period with continuous therapy for 5 months and subsequent intermittent therapy. Mean duration of therapy was 15 months.

### 3.3. Adverse Events

All patients with vismodegib (*n* = 30, 100%) and most patients with sonidegib (*n* = 11, 92%) had at least one treatment-related adverse event. Muscle spasms were the most common adverse event for both treatments, reported by 87% of patients (*n* = 26) for vismodegib and 67% (*n* = 8) for sonidegib. Alopecia was the second most common adverse event for both HhIs. Figure 3 shows a patient before treatment with vismodegib (1A) and after 13 months of therapy and complete hair loss (1B). Interestingly, weight loss was exclusively observed in patients over 60 years of age for both treatments.

73% of patients (*n* = 8) of the vismodegib group with continuous dosing schedule reported muscle spasms compared to 82% of patients (*n* = 9) with intermittent treatment (1–3 months on/off). 83% of patients (*n* = 6) with continuous sonidegib treatment reported muscle spasms versus 50% of patients (*n* = 6) with intermittent treatment. A remarkable difference between dosing schedules was observed for weight loss, which affected 45% of patients (*n* = 5) with continuous vismodegib treatment and only 18% of patients (*n* = 2) with intermittent treatment. Table 1 shows the frequency of adverse events for vismodegib, sonidegib and for different dosing regimens in our patient population, as well as results from the large clinical trials BOLT (sonidegib), ERIVANCE (vismodegib) and MIKIE (intermittent dosing of vismodegib) [12,13,14].

In order to better compare the influence of the dosing regimen on adverse events, we further analyzed the data of patients with both continuous and intermittent therapy with HhIs. Six patients from the vismodegib group had first continuous treatment (ranging from 5–9 months) followed by an intermittent treatment (1–3 months on/off, ranging from 12–58 months). All of these patients (*n* = 6, 100%) experienced an improvement of alopecia after switching the dosing regimen from continuous to intermittent (Table 2). Two patients reported an improvement of muscle spasms (*n* = 2, 33%) and one patient an improvement of dysgeusia (*n* = 1, 17%). One patient with sonidegib experienced significant weight loss under continuous therapy and achieved a stable weight under intermittent dosing.

By comparing the prevalence of adverse events between vismodegib and sonidegib, we found dysgeusia to be more frequent for vismodegib (*n* = 24, 80%) than sonidegib (*n* = 3, 30%). Alopecia, muscle spasms, weight loss and fatigue were also more frequent for vismodegib (Table 1). The analysis of adverse events for patients with both vismodegib and sonidegib subsequently (and same dosing regimen) showed a reduced occurrence of all assessed adverse events during treatment with sonidegib (Figure 2).

Regarding HhI-induced muscle spasms, nine patients were prescribed peroral chinine sulfate 250 mg once to twice a day and reported symptom improvement with regular intake. Two patients experienced a reduction in muscle spasms with the muscle relaxant tizanidine, and four patients with peroral magnesium.

### 3.4. Efficacy

#### 3.4.1. Vismodegib

##### Locally Advanced BCC

The best overall response in the group of laBCC (*n* = 9) was PR in four patients (44%) and SD in two patients (22%). Figure 3 shows a patient with a locally advanced BCC on the right cheek before treatment with vismodegib (2A) and after 6 months of treatment with a partial response (2B). Three patients (33%) were not evaluable due to not exactly measurable extent of the histologically confirmed tumor. The response at therapy stop was PR in three patients (33%), SD in three patients (22%), PD in one patient (11%), and three patients (33%) were not evaluable. Mean progression-free survival (PFS) was 13.5 months (range 2–28 months) and mean time to next treatment or death was 17.5 months (range 2–34 months).

##### Metastatic BCC

The best overall response in the group of mBCC (*n* = 5) was CR in one patient (20%), PR in two patients (40%) and PD in one patient (20%). Figure 3 shows a PET/CT imaging of a patient with mBCC prior to vismodegib treatment (3A) and after 3 months of therapy with a partial response (3B). One patient (20%) was not evaluable due to the not exactly measurable extent of the histologically confirmed tumor. The response at therapy stop was CR in one patient (20%), PD in three patients (60%), and one patient (20%) was not evaluable. Mean PFS was 9.3 months (range 3–19 months) and mean time to next treatment or death was 14.3 months (range 8–24 months).Multiple BCCs

The best overall response in the group of multiple BCCs (*n* = 16) was CR in nine patients (56%), and PR in four patients (25%). Three patients (19%) were not evaluable due to incomplete data in the medical history. The response at therapy stop (*n* = 14, two patients still ongoing) was CR in five patients (36%), PR in three patients (21%), PD in two patients (14%), and three patients (21%) were not evaluable.

According to our definition of progression in patients with multiple BCCs, only a limited number of patients were progressing on HhI therapy. Furthermore, the main treatment goal for these patients is a reduction in BCC excisions and thus improvement of quality of life. Consequently, we did not assess PFS as it remains an unsuitable endpoint in this patient population.

#### 3.4.2. Sonidegib

##### Locally Advanced BCC

The best overall response in this group (*n* = 5) was PR in two patients (40%), one of which had a combination therapy with itraconazole), SD in one patient (20%, therapy combined with itraconazole), and for two patients response was not evaluable due to unmeasurable tumor. The response at therapy stop (*n* = 4) was PD (*n* = 2, 50%), SD (*n* = 1, 25%), and one patient with non-evaluable response (25%).

Mean PFS was 12 months with one patient under combined therapy with sonidegib and itraconazole. Mean time to next treatment was 12 months (range 3–19 months).

##### Metastatic BCC

Only one patient in our cohort had metastatic BCC treated with sonidegib, while best overall response was PD (therapy combined with itraconazole); PFS was 3 months and time to next treatment was 11 months.

##### Multiple BCCs

The best overall response in this group (*n* = 6) was CR in five patients (83%) and PD in one patient (17%). The response at therapy stop was CR in three patients (75%) and PD in one patient (25%). Two patients were still on treatment. Mean time to next treatment was 18 months, but it has to be taken into consideration that most patients (*n* = 4, 67%) were still on ongoing treatment and had not yet switched to another treatment.

### 3.5. Therapy Discontinuation

#### 3.5.1. Vismodegib

Treatment was discontinued in 26 (87%) of 30 patients. The main reason for treatment discontinuation was adverse events (42%, *n* = 11). The majority of these patients discontinued due to muscle spasms (45%) and weight loss (45%) (Figure 1). Within the different subgroups of laBCC, mBCC and multiple BCCs, the main reason for treatment discontinuation was also adverse events for laBCC (55%) and multiple BCCs (31%) and PD for patients with mBCC (60%).

#### 3.5.2. Sonidegib

A total of 8 out of 12 patients with sonidegib discontinued treatment (67%), three of them due to PD (37.5%), three patients due to adverse events (37.5%), and two patients had no evidence of disease (25%).

### 3.6. Hedgehog Therapy Combined

Three patients with laBCC or mBCC had combined treatment with sonidegib and itraconazole pulsed therapy (2 weeks on/2 weeks off), and one patient had triple therapy with vismodegib, itraconazole pulsed therapy (2 weeks on/2 weeks off) and cetuximab, an EGFR inhibitor. Another patient had vismodegib and pulsed itraconazole. All these patients stopped therapy due to PD. Itraconazole is an antifungal agent with the ability to inhibit the Hedgehog signaling pathway by blocking the SMO receptor directly. We combined the different agents in patients with progressive disease or insufficient response under vismodegib and sonidegib to possibly overcome drug resistance [15].

## 4. Discussion

This retrospective analysis of 33 patients over an 8-year period provides an insight in the benefits and limitations of HhIs in patients with advanced BCC. The patient population is heterogeneous and distinctly different to patient populations included in the pivotal trials. Many patients of this study had considerable comorbidities, including hematologic malignancies and solid tumors. Concerning the distribution of the three different subgroups, 33% of patients had laBCC and 15% mBCC. Fifty-two percent of the patients suffered from multiple BCCs, in contrast to BOLT [13] and ERIVANCE [12] where only participants with laBCC and mBCC were included. This heterogeneity reflects the difficulties that we face in daily practice in patients with genetic syndromes or with multiple BCCs. Often patients present with a long history, and are unwilling to agree to further surgical procedures due to the associated inconveniences and welcome alternative treatment strategies. Overall, our data confirm the impressive antitumor efficacy of HhIs. Tumor regressions were observed in 88% of patients. In patients with underlying genetic syndromes, the frequency of surgical procedures was reduced substantially, which resulted in a major relief of disease burden.

Adverse events, mainly muscle spasms and significant weight loss, were the most frequent cause for treatment discontinuation overall in our study. This was observed to the same extent in both continuous and intermittent dosing regimens. Although the incidence of severe adverse events for HhIs is rare, the on-target effects of HhIs comprising dysgeusia, muscle spasms and alopecia are major issues for the patient even if they are graded 1 or 2 using Common Terminology Criteria for Adverse events (CTCAE). Therefore, good adverse-event management can improve medication adherence. This includes symptomatic medical therapy or an adapted dosing regimen. Muscle spasms could be subjectively reduced with quinine sulfate (200–250 mg twice a day). Some patients also benefited from peroral magnesium or muscle relaxants such as tizanidine. With a 2 months on/2 months off intermittent treatment we were able to avoid total alopecia, which was an important prerequisite for starting therapy, especially for women. We failed to influence dys- or hypogeusia. Even after discontinuation of therapy, recovery took months. Weight was assessed regularly in follow-ups, as HhI therapy must be discontinued at critical weight loss. Interestingly, weight loss was exclusively observed in patients over 60 years of age. We hypothesize that elderly people have a lower flexibility and adaptability in their eating habits to balance the effects of drug-induced alterations of taste including loss of appetite and aversion to certain foods, therefore leading to a nutritional deficiency. In the group of patients with BCNS, we noticed disappearance of palmoplantar pits under HhI therapy. Regression of jaw cysts under HhI treatment is described by Ally et al. [16] and was also observed in one of our patients with BCNS.

So far, one randomized double-blind prospective study (MIKIE) assessed safety and activity for long-term intermittent dosing of vismodegib for patients with multiple basal-cell carcinomas [14]. The results showed sustained treatment activity and tolerability with two different intermittent dosing regimens. However, a limitation of this study was the lack of a treatment group with continuous dosing, so that no direct comparison was possible. Interestingly, our data did not show a reduced incidence of adverse events comparing patients with continuous and intermittent therapy. This is probably due to a selection of patients experiencing several adverse events, mostly at the beginning of therapy, who were consequently managed with intermittent dosing to avoid early treatment discontinuation. By comparing adverse-event profiles in individual patients with both continuous and intermittent dosing regimens we observed an improvement of at least one adverse event per patient when changing to intermittent dosing (Table 2). In particular, alopecia and muscle spasms were markedly decreased in many patients. In the laBCC and mBCC group, the continuous dosing regimen was strictly reinforced in the initial months in contrast to patients with multiple BCCs, where an intermittent regimen with longer therapy breaks was applied. On the one hand, this was carried out for better tolerability and thus better compliance in view of the potentially lifelong treatment, especially in the context of genetic syndromes, and on the other hand, time to tumor regression was shorter in the group with smaller, but multiple BCCs than in the group with laBCC and mBCC. The limited patient numbers in each subgroup (laBCC, mBCC and multiple BCCs) and the uneven distribution of dosing regimens within these groups did not allow an adequate comparison of treatment responses between continuous and intermittent dosing.

Our data seem to suggest that sonidegib has less adverse events compared to vismodegib, which was especially observed in patients who were treated with both sonidegib and vismodegib sequentially (Figure 2). Therefore, switching from vismodegib to sonidegib in case of poor tolerability could be contemplated. Pharmacokinetic profiles of sonidegib have shown better tissue penetration and thus higher concentration of sonidegib in skin compared to vismodegib [17]. However, due to small patient numbers and shorter observational periods for sonidegib, these results must be interpreted with caution. Head-to-head studies for sonidegib and vismodegib are the only possibility to obtain reliable data on the tolerability.

Concerning second-line treatment in case of intolerance to or progression during a therapy with HhI for patients with laBCC or mBCC, approval by the FDA and EMA was granted for cemiplimab, a PD-1 antibody [18]. The approval was based on the results of an open-label, multicenter, single-arm phase 2 trial [19]. Response rate is lower than for treatment with HhI, but long response duration and possible cure in some patients—as we observe it in advanced squamous cell carcinoma—are attractive perspectives. We estimate that at least 10 patients reported in this paper are candidates for anti-PD1 therapy. There are also possible synergistic effects of simultaneous therapy with HhI and cemiplimab. Three of our patients switched after progression to an investigator-initiated trial (IIT) with treatment with cemiplimab and pulsed dosing of sonidegib [20].

By managing this challenging patient population, we noticed that there are a number of patients with laBCC who presented to a physician very late. Their psychology was intriguingly determined by disease denial. We attributed this to a certain habituation to the painless process associated with skepticism toward medical procedures and personnel. Interestingly, long-term intensive medical care had a positive impact on this attitude reflected by good adherence to the treatment plan and compliance concerning medications and visits to the hospital.

The limitations of this study include inconsistent documentation of medical history to accurately classify adverse events according to CTCAE grading. In addition, the comparison of tumor response within the groups was difficult due to the lack of systematic predefined measurement endpoints and because some patients had a previous HhI therapy within a clinical trial or had switched from vismodegib to sonidegib.

The strength of this study is the personalized continuous care of a patient population with many comorbidities and genetic syndromes and individual therapy adjustments with breaks in therapy based on the patient’s needs, which improves compliance and may ameliorate adverse events.

## 5. Conclusions

This real-world data set illustrates how intermittent dosing and concomitant therapies prolong the benefits of HhIs in a difficult patient population of advanced age and significant comorbidities. It also provides important information on the characteristics of patients who need additional treatment options in the future.

## Figures and Tables

**Figure 1 cancers-14-02496-f001:**
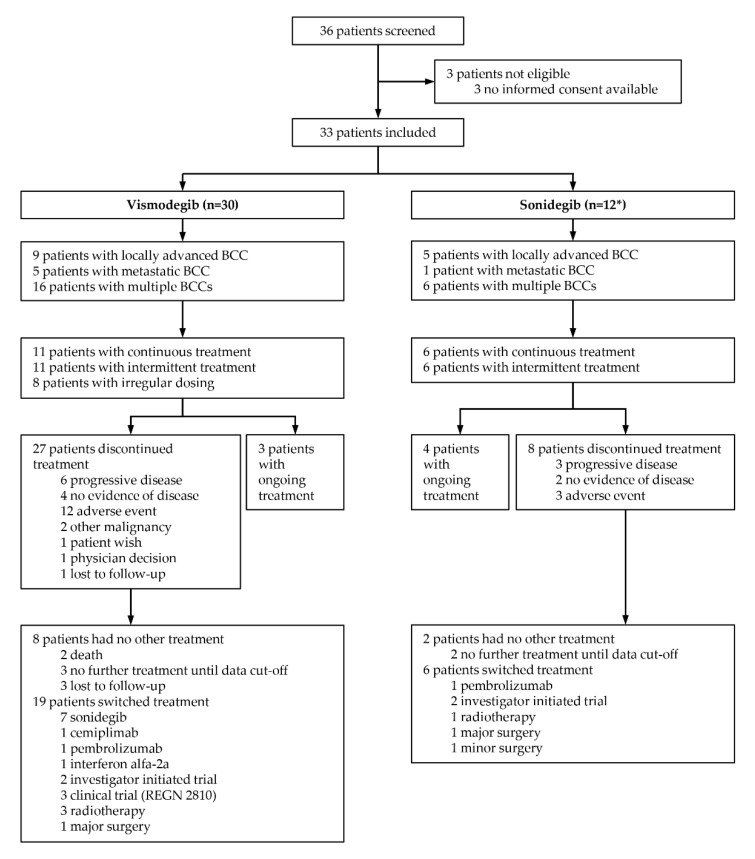
Patient distribution for the two hedgehog inhibitors vismodegib and sonidegib, treatment modality and follow-up therapy. * 3 patients started with sonidegib as the first hedgehog inhibitor therapy outside a clinical trial and 9 patients were switched from vismodegib to sonidegib: *n* = 3 + 9 = 12.

**Figure 2 cancers-14-02496-f002:**
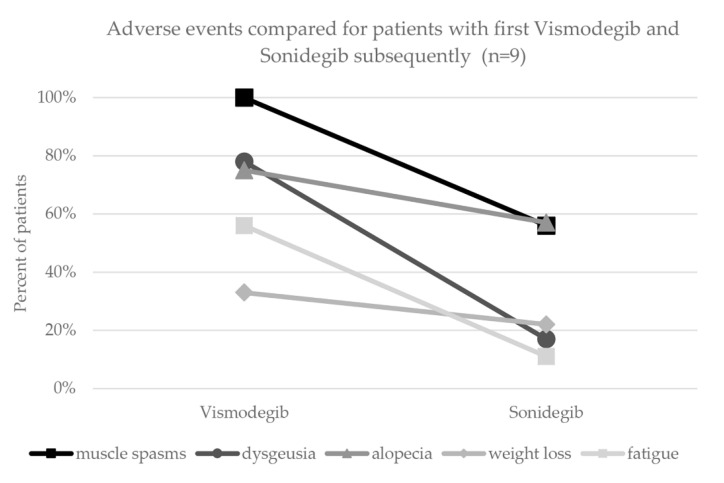
Comparison of adverse events for vismodegib versus sonidegib in 9 patients with both treatments subsequently.

**Figure 3 cancers-14-02496-f003:**
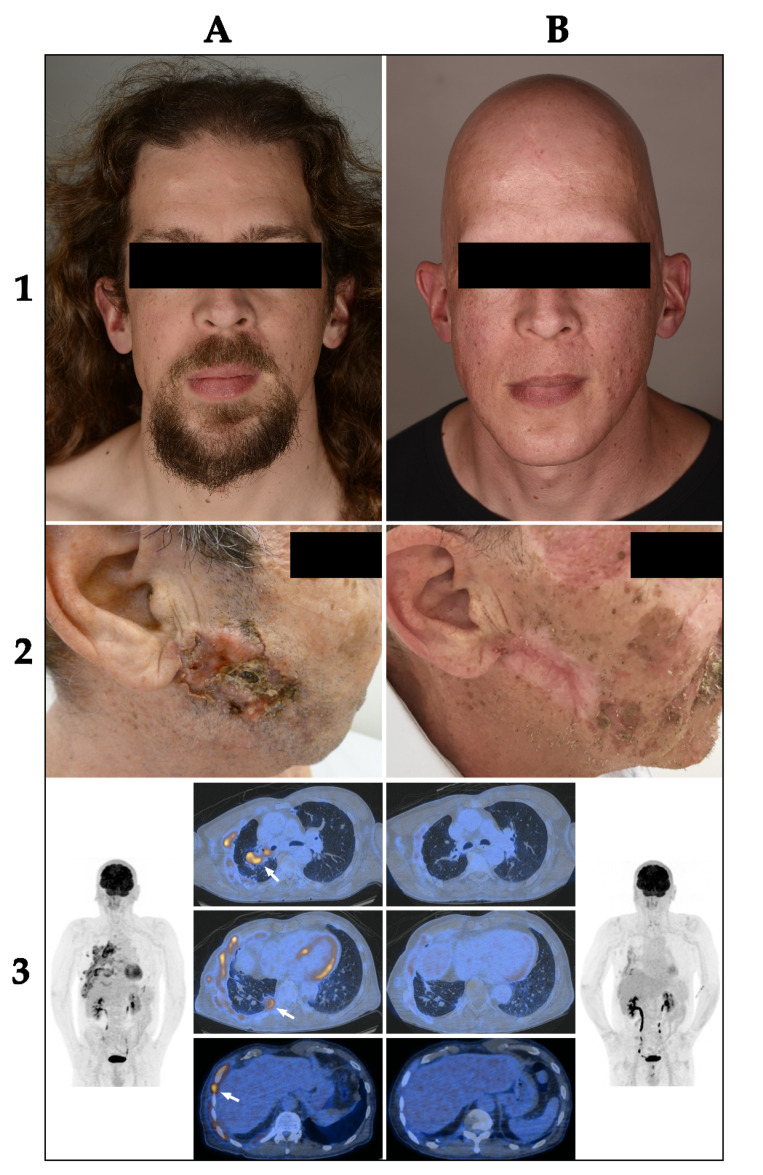
Treatment response and adverse events of HhIs (**1**): Patient before treatment (**1A**) and after 13 months of therapy with vismodegib with complete hair loss (**1B**). (**2**): Locally advanced BCC on the right cheek before (**2A**) and after 6 months of treatment with vismodegib (**2B**). (**3**): 18F-FDG-PET/CT imaging prior to HhI treatment in 06/2017 (**3A**) and after 3 months of therapy with vismodegib in 09/2017 (**3B**). Several pulmonary and pleural metastases (arrows) demonstrated complete metabolic response in the follow-up scan in 09/2017. (**1**–**3**) are all different patients.

**Table 1 cancers-14-02496-t001:** Adverse events (AEs) reported for different HhIs, different dosing regimens and for comparison of AEs reported in key trials.

Adverse Events (Any Grade)	Vismodegib Continuous (*n* = 11)	Vismodegib Intermittent (*n* = 11)	Vismodegib All (*n* = 30)	Sonidegib Continuous (*n* = 6)	Sonidegib Intermittent (*n* = 6)	Sonidegib All (*n* = 12)	BOLT [13] (*n* = 79)	ERIVANCE [12] (*n* = 104)	MIKIE [14] Group A (*n* = 114)	MIKIE [14] Group B (*n* = 113)
alopecia	55%	64%	82%	80%	40%	60%	50%	68%	63%	65%
muscle spasms	73%	82%	87%	83%	50%	67%	52%	74%	73%	83%
dysgeusia	73%	82%	80%	20%	25%	30%	41%	56%	78%	80%
weight loss	45%	18%	47%	50%	33%	42%	29%	52%	21%	19%
fatigue	18%	27%	47%	17%	0%	8%	29%	42%	21%	23%

**Table 2 cancers-14-02496-t002:** Improvement of adverse events after switch from continuous to intermittent dosing for patients with vismodegib or sonidegib.

			Improvement of Adverse Event				
Patient	HhI	Dosing Regimen	Alopecia	Muscle Spasms	Dysgeusia	Weight Loss	Fatigue
Patient I	Vismodegib	5 months continuous, 12 months intermittent	x			n.a.	
Patient II	Vismodegib	5 months continuous, 29 months intermittent	x	x		n.a.	n.a.
Patient III	Vismodegib	5 months continuous, 19 months intermittent	x			n.a.	n.a.
Patient IV	Vismodegib	6 months continuous, 16 months intermittent	x				x
Patient V	Vismodegib	7 months continuous, 8 months intermittent	x	n.a.	x	x	n.a.
Patient VI	Vismodegib	9 months continuous, 58 months intermittent	x	x		n.a.	x
Patient VII	Sonidegib	5 months continuous, 11 months intermittent				x	

n.a. = not applicable; patient did not experience this AE with either continuous or intermittent therapy.

## Data Availability

Not applicable.

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
