# Peer review of "Eight Years of Real-Life Experience with Smoothened Inhibitors in a Swiss Tertiary Skin Referral Center"

_cancers, 2022, doi:10.3390/cancers14102496_

Round 1
Reviewer 1 Report
The article gives important insights in a real world manner that is very important, especially for physisians. Althought such studies may have some limitations, they are very wellcome as an experience exchange between Clinicians.
Author Response
Thank you for your time you spent carefully reviewing the manuscript and the valuable comments.
The manuscript was corrected for english language and minor spell checks by a native-speaker.
Reviewer 2 Report
This is an interesting study and the authors have collected a unique dataset using cutting edge methodology. The paper is generally well written and structured.
Here are some minor suggestions:
Did you assess treatment response only clinically ? Did you do dermoscopy to assess treatment response ? did you confirm it with histopathology examination ?
Did you do the statistical analysis of the date ?
In my opinion the topic is original and relevant in the field because in the study authors present long-term experience in treatment with HhIs. They present and compare effectiveness of treatment and its adverse events in three groups (laBCC, mBCC and multiple BCC), in previous study patients with multipleBCC were not included to the study.
- Three groups patients in one study: laBCC multiple BCC and mBCC
- Heterogenous group of patients
- Comparison continuous and intermittent therapy
In my opinion the statistical analysis of the obtained date would be valuable
the references appropriate references 1,2 3,4 could be from more recent studies
I don’t have any additional comments on the table and figures, only additional clinical pictures can be added
Author Response
Thank you for your time you spent carefully reviewing the manuscript and the valuable comments. In what follows the referees’ comments are in black and the authors’ responses are in red.
Did you assess treatment response only clinically? Did you do dermoscopy to assess treatment response? Did you confirm it with histopathology examination?
We assessed treatment response clinically, including dermoscopy, for all patients. Histopathological confirmation of treatment response was not done routinely.
We agree that this should be specified in the text and have updated the manuscript as follows:
We further obtained data on treatment response, which was clinically assessed by the investigator, including dermoscopy, according to the patient’s medical record in laBCC, and according to the RECIST criteria in patients with mBCC. For patients with multiple BCCs, response was also assessed clinically including dermoscopy.
Did you do statistical analysis of the data?
The patient population in this study is very heterogenous regarding previous treatments and indication criteria. Especially, an efficacy analysis would have to be done for each indication subgroup (laBCC, mBCC and multiple BCCs) separately. In our study population, a substantial part of the patients received a previous therapy with a hedgehog inhibitor within a clinical trial (ERIVANCE or BOLT) or received Vismodegib and Sonidegib subsequently. This did not allow us a direct comparison of tumor response of patients from other studies who have had a first-line therapy with a hedgehog inhibitor. Moreover, the tumor response of many patients was not evaluable due to not exactly measurable extent (clinically and radiologically) of the histologically confirmed tumor(s) or due to the combination of the hedgehog inhibitor therapy with other tumor therapies (such as immunotherapy, chemotherapy etc). By excluding all these patients, the number of patients in each subgroup (laBCC, mBCC and multiple BCCs) is too small for an efficacy analysis. Moreover, adverse events were not graded following CTCAE, impeding comparison with other studies.
Overall, a rigid statistical analysis of this patient collection is not possible.
Reviewer 3 Report
This manuscript reported the clinical results of hedgehog inhibitors vismodegib and sonidegib in treating advanced basal cell carcinoma. With 33 patients, the pitfalls and benefits of vismodegib or sonidegib was showed. Although this manuscript demonstrated valuable clinical observation, more insight for using hedgehog inhibitors to treat advanced basal cell carcinoma can be provided by comparing with other clinical results regarding vismodegib and sonidegib, and disussing whether the outcomes in this manuscript is population-specific or cancer-type specific.
Author Response
Thank you for your time you spent carefully reviewing the manuscript and the valuable comments. In what follows the referees’ comments are in black and the authors’ responses are in red.
Although this manuscript demonstrated valuable clinical observation, more insight for using hedgehog inhibitors to treat advanced basal cell carcinoma can be provided by comparing with other clinical results regarding vismodegib and sonidegib, and disussing whether the outcomes in this manuscript is population-specific or cancer-type specific
The patient population in this study is very heterogenous regarding previous treatments and indication criteria. Especially an efficacy analysis would have to be done for each indication subgroup (laBCC, mBCC and multiple BCCs) separately. In our study population a substantial part of the patients received a previous therapy with a hedgehog inhibitor within a clinical trial (ERIVANCE or BOLT) or received Vismodegib and Sonidgebig subsequently. This did not allow us a direct comparison of tumor response of patients from other studies who have had a first-line therapy with a hedgehog inhibitor. Moreover, the tumor response of many patients was not evaluable due to not exactly measurable extent (clinically and radiologically) of the histologically confirmed tumor(s) or due to the combination of the hedgehog inhibitor therapy with other tumor therapies (such as immunotherapy, chemotherapy etc). By excluding all these patients, the number of patients in each subgroup (laBCC, mBCC and multiple BCCs) is too small for an efficacy analysis.
Incidence of treatment related adverse events of Vismodegib and Sonidegib in our study was similar to that in other retrospective real-life studies [1,2]. A decrease of adverse event severity after a change from continuous to intermittent therapy for Vismodegib and Sonidegib (as we showed in our study) could also be shown in other retrospective real-life studies/case series, but –in contrast to our study- these studies did not specify which adverse event in particular (muscle cramps, dysgeusia, fatigue, alopecia and weight loss) improved [3-5]. Furthermore until now there is few data from real-life (outside clinical trials) on adverse events under vismodegib compared to sonidegib for patients having had both treatments subsequently.
Overall our results regarding treatment response and adverse events are similar to those from large prospective clinical trials and other real-life studies, but comparison is limited due to the above mentioned reasons.
- Verkouteren, B.J.A.; Wakkee, M.; Reyners, A.K.L.; Nelemans, P.; Aarts, M.J.B.; Rácz, E.; Terra, J.B.; Devriese, L.A.; Alers, R.J.; Kapiteijn, E.; et al. Eight years of experience with vismodegib for advanced and multiple basal cell carcinoma patients in the Netherlands: a retrospective cohort study. Br J Cancer 2021, 124, 1199-1206, doi:10.1038/s41416-020-01220-w.
- Villani, A.; Fabbrocini, G.; Costa, C.; Scalvenzi, M. Sonidegib efficacy and tolerability in advanced basal cell carcinoma: A single-center real-life experience. J Am Acad Dermatol 2022, 86, e175, doi:10.1016/j.jaad.2021.11.041.
- Woltsche, N.; Pichler, N.; Wolf, I.; Di Meo, N.; Zalaudek, I. Managing adverse effects by dose reduction during routine treatment of locally advanced basal cell carcinoma with the hedgehog inhibitor vismodegib: a single centre experience. J Eur Acad Dermatol Venereol 2019, 33, e144-e145, doi:10.1111/jdv.15367.
- Scalvenzi, M.; Costa, C.; Cappello, M.; Villani, A. Reply to Woltsche N. et al. Managing adverse effects by dose reduction during routine treatment of locally advanced basal cell carcinoma with the hedgehog inhibitor vismodegib: a single-centre experience. J Eur Acad Dermatol Venereol 2019, 33, e145-e147, doi:10.1111/jdv.15469.
- Toffoli, L.; Conforti, C.; Zelin, E.; Vezzoni, R.; Agozzino, M.; di Meo, N.; Zalaudek, I. Locally advanced basal cell carcinoma: Real-life data with sonidegib. Dermatol Ther 2022, e15441, doi:10.1111/dth.15441.